# Production, Kinetic/Thermodynamic Study, and Evaluation of the Influence of Static Magnetic Field on Kinetic Parameters of β-Fructofuranosidase from *Aspergillus tamarii* Kita UCP 1279 Produced by Solid-State Fermentation

**DOI:** 10.3390/biotech12010021

**Published:** 2023-03-03

**Authors:** Rodrigo Lira de Oliveira, Aldeci França Araújo dos Santos, Bianca Alencar Cardoso, Thayanne Samille da Silva Santos, Galba Maria de Campos-Takaki, Tatiana Souza Porto, Camila Souza Porto

**Affiliations:** 1School of Food Engineering, Federal University of Agreste of Pernambuco/UFAPE, Av. Bom Pastor, Boa Vista, s/n, Garanhuns 55296-901, Brazil; 2Education Unit of Penedo, Federal University of Alagoas/UFAL, Avenida Beira Rio, s/n, Penedo 57200-000, Brazil; 3Nucleus of Research in Environmental Sciences and Biotechnology, Catholic University of Pernambuco/UNICAP, Rua do Príncipe, 526, Recife 50050-590, Brazil; 4Department of Morphology and Animal Physiology, Federal Rural University of Pernambuco/UFRPE, Av. Dom Manoel de Medeiros, s/n, Recife 52171-900, Brazil

**Keywords:** *Aspergillus*, β-fructofuranosidase, kinetic, solid-state fermentation, static magnetic field, thermodynamics

## Abstract

β-fructofuranosidases (FFases) are enzymes involved in sucrose hydrolysis and can be used in the production of invert sugar and fructo-oligosaccharides (FOS). This last is an important prebiotic extensively used in the food industry. In the present study, the FFase production by *Aspergillus tamarii* Kita UCP 1279 was assessed by solid-state fermentation using a mixture of wheat and soy brans as substrate. The FFase presents optimum pH and temperature at 5.0–7.0 and 60 °C, respectively. According to the kinetic/thermodynamic study, the FFase was relatively stable at 50 °C, a temperature frequently used in industrial FOS synthesis, using sucrose as substrate, evidenced by the parameters half-life (115.52 min) and *D*-value (383.76 min) and confirmed by thermodynamic parameters evaluated. The influence of static magnetic field with a 1450 G magnetic flux density presented a positive impact on FFase kinetic parameters evidenced by an increase of affinity of enzyme by substrate after exposition, observed by a decrease of 149.70 to 81.73 mM on *K_m_*. The results obtained indicate that FFases present suitable characteristics for further use in food industry applications. Moreover, the positive influence of a magnetic field is an indicator for further developments of bioprocesses with the presence of a magnetic field.

## 1. Introduction

Fructo-oligosaccharides (FOS) are a mixture of fructo-oligomers with two or three fructose units bound to the β-(2,1) or β-(2,6) position of sucrose. These oligosaccharides are broadly used in the food industry due their technological and functional properties, especially in the production of food formulations with low or no added sugars. FOS are naturally present in vegetables, cereals and fruits or can be obtained by enzymatic synthesis by inulin hydrolysis or transfructosylation of sucrose residues [1,2]. FOS-forming enzymes that use sucrose as a substrate are classified into two main groups: fructosyltransferases (FTase, EC 2.4.1.9) and β-fructofuranosidases (FFase, EC 3.2.1.26). These last act mainly on sucrose hydrolysis, obtaining as a product an equimolar mixture of fructose and glucose, commonly called invert sugar. In FOS synthesis, the FFases acts according to two mechanisms: reverse hydrolysis and transglycosylation; higher FOS yields by the action of this enzyme are obtained by applying high sucrose concentrations [3,4].

FFases can be obtained from vegetable and microbial sources, this last being preferentially used. Filamentous fungi belonging the *Aspergillus*, *Aureobasidium* and *Penicilium* genera produce enzymes with the best yields of FOS production [5]. The *Aspergillus tamarii* strains stand out in the production of both FTases and FFases, intra and extracellularly [6,7,8]. In a previous study, the *A. tamarii* Kita UCP 1279 strain was able to produce FFases capable of being employed in FOS synthesis [9]. Conventionally, the production of microbial enzymes, including the FFases, can be carried out by submerged (SmF) or solid-state fermentation (SSF). This last is characterized as a process in which microorganisms grow in a medium with a very low water content or with absence of free water. The SSF simulates the microorganism habitat, especially for filamentous fungi, and stimulates the synthesis of considerable amounts of enzymes [10]. An advantage of SSF in relation to SmF is the wide use of solid agro-industrial wastes as substrates, which are very abundant and generally underutilized. Among the agro-industrial wastes and co-products that could be exploited for FOS and FOS-forming enzymes production are sucrose-rich solutions, leaves, fruit peels, some bagasse, brans and coffee processing by-products [11].

A relevant step to measure FFase performance and predict their suitable use in specific applications is biochemical characterization, in terms of temperature and pH, and the kinetic/thermodynamic study. This last approach provides a better understanding of the extent of enzymatic reaction and the enzyme thermostability at a given operating temperature, which is essential to the prediction of success of the bioprocess [12]. Moreover, the study of techniques for enhancement in the stability and enzyme catalytic potential have gained great interest. The utilization of physical factors such as electromagnetic field and magnetic field to stimulate and modify the enzyme activity and its catalytic properties is an interesting approach. The application of an external magnetic field may influence the molecular structure of a given enzyme, thus modifying its catalytic properties [13]. Some authors observed a positive impact of exposure to the enzyme on magnetic fields on enzyme kinetics [14,15], evidenced by an increase of affinity to correspondent substrate, which can favor several bioprocesses. The chemical or physical modifications in the enzymatic structural modification after magnetic flux have been studied in different industrial interest enzymes, such as lipases [16] and amylases [14], and clinical interest enzymes, such as fibrinolytic proteases [17]. However, to best of our knowledge, no studies exist evaluating this influence on FOS forming enzymes.

On basis of this background, the aims of the present study were to (1) evaluate the improvement of the production of FFase from *Aspergillus tamarii* Kita UCP 1279 through solid-state fermentation using mixtures of agro-industrial substrates, (2) characterize the enzyme in terms of operational conditions (pH and temperature) and kinetic/thermodynamic parameters for enzyme thermal denaturation and (3) evaluate the impact of the exposure of a static magnetic field on kinetic constants (*K_m_* and *V_max_*) of FFase.

## 2. Materials and Methods

### 2.1. Culture Maintenance and Preparation of Spore Suspension

The fungal strain *Aspergillus tamarii* Kita UCP 1279, isolated from Caatinga soil, located in the Northeast of Brazil, and provided by culture collection of Catholic University of Pernambuco (UNICAP) (Recife, PE, Brazil), was used in the present study. The strain was maintained in Czapek Dox Agar medium and preserved in mineral oil at 25 ± 1 °C (room temperature). To produce fungal spores, the strain was inoculated in Potato Dextrose Agar (PDA) medium and incubated for 7 days at 30 °C. A NaCl solution (0.9%, *w v*^−1^) added with Tween 80 (0.01%, *w v*^−1^) was used to suspend the spores. Finally, the spore suspension used on the solid-state fermentation process was adjusted to 10^7^ spores g^−1^ of substrate.

### 2.2. Solid-State Fermentation Conditions for FFase Production

The FFase production by SSF was performed in 125 mL Erlenmeyer flasks using mixtures of wheat and soy brans (0.5–2.0 mm) as substrate for 72 h at 30 °C. The SSF was conducted with an initial moisture of the substrate of 60%, adjusted with a nutritive solution composed of sucrose (20 %, *w v*^−1^) and yeast extract (0.5 %, *w v*^−1^) at pH 5.0, prepared in acetate buffer (0.1 M), and spore suspension, prepared as described in Item 2.1. After the end of SSF final, the enzymatic extract was obtained using 7.5 mL of distilled water per g of substrate, then it was subjected to orbital stirring at 130 rpm for 90 min. Finally, the enzymatic extract was filtered through cheesecloth and the solids were removed by centrifugation at 3000× *g* for 20 min and 4 °C (Thermo Fisher Scientific-ST16R, Karlsruhe, Germany). Finally, the obtained supernatant was stored at −22 °C for later analytical determination.

The influence of variables’ substrate amount (3, 5 and 7 g), soy bran proportion (20, 50 and 80%) and sucrose concentration (15, 20 and 25%) on FFase production were studied according to a 2^3^ full factorial design plus four central points. The Statistica 7.0 software package (Statsoft Inc., Tulsa, OK, USA) was used for performing the statistical analysis of the experimental results obtained from the factorial design.

### 2.3. FFase Hydrolytic Activity

FFase hydrolytic activity was performed according to the conditions described to Sangeetha et al. [18] with modifications, using a sucrose as substrate (60%, *w v*^−1^; 0.1 M; pH 5.0) preparate in sodium acetate buffer and the reaction was performed by incubation of the enzymatic extract and substrate at 55 °C for 1 h. Then, the reaction was stopped by the rapid cooling and the reactional mixture was evaluated according to the glucose release using a glucose oxidase colorimetric kit (Liquiform, Labtest S. A., Lagoa Santa, MG, Brazil). The unit of FFase activity was defined as the amount of enzyme needed to release 1 μmol glucose per minute under conditions described.

### 2.4. Optimum Reaction PH and Temperature of Sucrose Hydrolysis Catalyzed by FFase

The optimal temperature of FFase activity was evaluated by activity tests carried out at different temperatures (20–70 °C), varying every 10 °C and maintaining constant pH (5.0), while pH effect on FFase activity was studied by activity tests performed with the substrate (sucrose) diluted in different buffers (0.1 M): citrate (pH 3.0–4.0), acetate (pH 4.0–5.0), citrate–phosphate (pH 5.0–7.0) and Tris HCl (pH 7.0–9.0) at constant temperature (55 °C). Tukey’s test was used to verify statistically significant differences (*p* < 0.05) among samples.

### 2.5. Kinetics and Thermodynamics of Thermal Denaturation of FFase

The FFase thermal denaturation was evaluated at 50–65 °C by residual activity experiments during a maximum time interval of 180 min. Taking account of the denaturation process can be represented as a temperature-dependent first-order reaction and represented in logarithm form by Equation (1); the deactivation rate constant (*k_d_*) can be estimated by plotting the experimental data of ln*ψ* versus time. *ψ* is the residual activity coefficient, calculated by the ratio of the residual enzyme activity (*A*) to that at the beginning of thermal treatment (*A*_0_).
(1)$$\ln\psi =-k_{d}t$$
where *t* is the time (min).

From the *k_d_* values, the calculated parameters half-life (*t*_1/2_) and decimal reduction time (*D*-value) were calculated according to Equations (2) and (3). The *t*_1/2_ is defined as the time after which the enzyme activity decreased to 50% of the initial value, while *D*-value corresponds to the time needed for a 10-fold reduction of the initial activity.
(2)$$t_{1/2}=\frac{\ln2}{k_{d}}$$
(3)$$D=\frac{\ln10}{k_{d}}$$

The *D*-value is often accompanied by another parameter, the *Z*-value, which is defined as the temperature increase needed to achieve a 10-fold reduction in the *D*-value and was estimated from the slope of the plot of log*D* versus *T* (°C).

The activation energy of enzyme denaturation (*E***_d_*) was estimated from the slope of the straight line of an Arrhenius-type plot (ln*k_d_* vs. 1/T). The thermodynamic parameters enthalpy (∆*H***_d_*), Gibbs free energy (∆*G***_d_*) and entropy (∆*S***_d_*) related to FFase thermal denaturation were calculated according to Equations (4)–(6), as follow:
(4)$$\Delta H_{d}^{*}=E_{d}^{*}-RT$$
(5)$$\Delta G_{d}^{*}=-RT\ln\left(\frac{k_{d}h}{k_{b}T}\right)$$
(6)$$\Delta S_{d}^{*}=\frac{\Delta H^{*}-\Delta G^{*}}{T}$$
where *h* is the Planck constant (6.626 × 10^−34^ J s), *k_b_* the Boltzmann (1.381 × 10^−23^ J K^−1^) constant and *T* is the reference temperature (298.15 K).

### 2.6. Effect of Static Magnetic Field of Kinetic Parameters of FFase Catalyzed Reaction

The Michaelis constant (*K_m_*) and maximum reaction rate (*V_max_*) were calculated using the Lineweaver–Burk plot of FFase activity versus sucrose concentrations (25 ≤ *S*_0_ ≤ 700 mg mL^−1^). The influence of static magnetic flux (SMF) on kinetic parameters was investigated using two neodymium magnets (4.0 cm × 0.2 cm × 0.5 cm) generating a magnetic field with 1450 G (0.145 T) magnetic flux density. Magnets were maintained at standardized distances using the apparatus described in Figure 1 during the FFase activity determination.

## 3. Results and Discussion

### 3.1. Production of FFase by Solid-State Fermentation

In a previous study involving FOS-forming enzyme production by *A. tamarii*, Kita was evaluated using only wheat bran as substrate and the maximum production observed was 62.47 U mL^−1^ [8]. However, studies of our research group verified a positive effect of soy bran on FFase production by SSF [7], and on basis of this indication, the production of this enzyme by *A. tamarii* Kita UCP 1279 was investigated using different mixtures of wheat and soy bran and other variables according to a 2^3^ full factorial design. The results obtained and experimental conditions of the experimental design are presented in Table 1; the maximum production (204.31 U mL^−1^) was observed at conditions described in run 6 (3 g of substrate, 80% of soy bran and 15% of sucrose). The result obtained was approximately 3.3-fold higher than that obtained using the medium composed only of wheat bran, confirming the positive influence of this component. Similar qualitative behavior was reported by Rustiguel et al. [19] who evaluated the FFase production from *A. phoenicis* under SSF and reported an increase of 16.05 on enzyme production in a binary equal mixture with wheat and soy brans in relation to cultivation using only wheat bran.

The statistical analysis was performed and was expressed in Pareto chart (Figure 2) and it was observed that all independent variables present a significant effect on FFase production. The substrate amount presents a positive effect on FFase production, on the other side, the proportion of soy bran and sucrose concentration presents a negative effect. The substrate amount can be related to the aeration of the fermentative system, and higher substrate amounts are related to lower O_2_ available for aerobic processes. Despite the proved positive impact of the addition of soy bran to SSF cultivation medium, it was observed that this variable presented a negative effect, indicating that lower percentages of this component enable improvement in enzyme production. As is known, sucrose is considered the best carbon source to produce enzymes with transfructosylating activity [20]. However, it was observed that lower levels of this variable can obtain higher levels of FFase production. The tertiary interaction of all variables evaluated present an antagonistic, statistically significant interaction.

### 3.2. Effect of PH and Temperature on Sucrose Hydrolysis Catalyzed by FFase

Environmental factors such as pH and temperature can influence the rate of enzyme-catalyzed reactions through reversible or irreversible modifications in the enzyme structure [21]. Therefore, the definition of optimum pH and temperature profile provides valuable information for the utilization of enzymes in bioprocesses. After the determination of fermentative conditions, the FFase was characterized in terms of pH and temperature and it was observed that enzymes present maximum activity in a pH range of 5–7 and 60 °C, as observed in Figure 3. Dapper et al. [22] report an optimum pH at 6.0 for *A. versicolor* FFase, on the other side, Choukade and Kango [6] report a maximum FTase activity at pH 7.0 for mycelial *Aspergillus tamarii* enzyme. Considering the temperature influence, the optimum FTase activity was observed at 60 °C, and at 70 °C, the activity was lower, presenting 17.47% relative activity. Similar optimum temperatures were also reported by Xu et al. [23] and Smaali et al. [24] for FFases obtained by *Penicillium oxalicum* and *Aspergillus awamori*, respectively.

### 3.3. Kinetic and Thermodynamics of FFase Thermal Inactivation

The FFase thermostability was evaluated in terms of kinetic and thermodynamic parameters for FFase thermal denaturation. For this purpose, experiments of FFase residual activity were performed at a temperature range of 50–65 °C. The results obtained were expressed in terms of residual activity coefficient (*ψ*) and were plotted to semi-log plots (Figure 4) for estimation of *k_d_* values; the results for this constant were estimated with suitable correlation (0.971 ≤ *R*^2^ ≤ 0.995) and presented in Table 2.

From the *k_d_* values, the kinetic parameters half-life (2.16 ≤ *t*_1/2_ ≤ 115.52 min) and decimal reduction time were calculated (7.19 ≤ *D*-value ≤ 383.76 min). These results indicated that the relative stability of *A. tamarii* Kita FFase presents relative stability at 50 °C, the temperature generally used in FOS synthesis at the industrial level using sucrose as substrate [25]. As observed, the *A. tamarii* Kita FFase presents a typical decay of first-order inactivation process, being the increase in *k_d_* values, and consequently decrease of *t*_1/2_ and *D*-values. The denaturation was more evident for temperatures higher than 55 °C, indicating that the enzyme is not thermostable in this temperature range. The *D*-value is usually accompanied by the thermal resistance constant (Z-value), which corresponds to the temperature rise necessary for the thermal denaturation curve to complete a logarithmic cycle and reduce the *D*-value to *D*/10 [26]. A Z-value result of 7.61 °C was obtained from the slope of plot of log*D* versus T (°C) (data not shown) with satisfactory correlation (*R*^2^ = 0.88). Low values for this parameter, as obtained in the present study, mean more sensitivity to rise in temperature than to duration of heat treatment [27]. A similar quantitative result (6.81 °C) was obtained by de Oliveira et al. [12] for FFase obtained for a different *A. tamarii* strain.

The thermodynamic parameters corresponding to FFase thermal deactivation were also calculated. This included the activation energy of enzyme thermal denaturation (*E*_d_*) estimated according to the slope of the Arrhenius-type plot (Figure 5) obtained by plotting ln*k_d_* versus the reciprocal absolute temperature (1/T) and a particular elevated value for this parameter was obtained (274.29 kJ mol^−1^). The parameters Δ*H***_d_*, Δ*S***_d_* and Δ*G***_d_* were calculated for each temperature evaluated and were given in Table 2.

The activation energy (*E*_d_*) is a relevant parameter for evaluation of enzyme thermostability and means the minimum amount of energy needed for the deactivation process. Higher values for this parameter are indicative of stability, as observed for FFase from *A. tamarii* Kita (274.29 kJ mol^−1^). A similar result was observed by de Oliveira et al. [10] for FFase from *A. tamarii* (293.08 kJ mol^−1^). The *E**_*d*_ is directly related to the Δ*H**_*d*_ and for this parameter, results between 271.60–271.48 kJ mol^−1^ were obtained. This parameter is related to disruption of non-covalent bonds, including hydrophobic interactions and, as is known, the energy needed to remove a -CH_2_ moiety from a hydrophobic linkage is 5.4 kJ mol^−1^ [28]. Then, it is possible to estimate the amount of broken non-covalent bonds for the denaturation process to occur. In this case, for FFase denaturation, the rupture, as an average, of 50.3 bonds is needed.

The opening-up of the protein structure resulting from the FFase denaturation process is accompanied by a rise in degree of disorder and system randomness, evidenced by large and positive Δ*S***_d_* values [29], as observed in the present study (508.55–522.36 J K^−1^ mol^−1^). The Δ*G***_d_* incorporates both the enthalpy and entropy contributions, being the most suitable parameter to evaluate the enzyme thermal denaturation phenomenon and, consequently, the biocatalyst stability [30]. Small or negative Δ*G***_d_* values are associated with a more spontaneous process. On the other side, positive values, as obtained in the present study (97.50–104.68 kJ mol^−1^), indicate resistance to denaturation. Then, the lower Δ*G***_d_* values obtained for temperatures of 60 and 65 °C confirm the decreased thermostability of FFase in this temperature range, indicated by the parameters *t*_1/2_ and *D*-values.

### 3.4. Influence of Static Magnetic Field on Kinetic Constants of FFase for Sucrose Hydrolysis

The kinetic parameters related to the catalyzed reaction (*K_m_* and *V_max_*) were investigated in the presence and absence of a magnetic field. To best of our knowledge, this is the first report of this approach on the kinetic study of FOS-forming enzymes and can contribute to the understanding of the impact of a magnetic field on the structural and catalytic properties of the enzyme. The results of both experimental conditions were adjusted with suitable correlation (0.941 ≤ *R*^2^ ≤ 0.994) to the Lineweaver Burk plots, as observed in Figure 6. The *K_m_* values were 149.70 and 81.73 mM without and with magnetic field, respectively, and a decrease in *V_max_* with the exposition of magnetic field (86.20 to 76.92 mM min^−1^) was also observed.

The presence of static magnetic flux (SMF) can have modifications and influences on the structure and stability of biomolecules. In the present study a decrease of 45% on *K_m_* in experiments conducted in presence of SMF was observed, indicating an increase in affinity of the enzyme by substrate. On the other hand, a decrease of approximately 11% in *V_max_* was observed. Similar qualitative behavior for *K_m_* was observed by Liu et al. [14] when the immobilized α-amylase kinetic parameters in the presence of 0.15 T were evaluated. However, other authors, such as Albuquerque et al. [17], did not observe significant differences in *K_m_* and *V_max_* for fibrinolytic protease from *Mucor subtilissimus* and a negative impact of SMF was evidenced by increase in *K_m_* and decrease in *V_max_,* as reported by Sun et al. [31] for a vegetable pectinase. Among the different mechanisms involved in modifications of biocatalyst activity by exposure to SMF, the radical pair recombination has been the preferred explanation, though it requires conditions at moderate magnetic fields to be effective [17,32], which is the case of the present experiment. The differences in the kinetic parameters of FFase would suggest modifications in the protein structure influenced by an SMF. These structural modifications have already been observed by Fraga et al. [16], especially on tertiary structure, for commercial lipase Eversa^®^ Transform 2.0. The positive impact on kinetic parameters obtained in the present study can be a starting point for studies involving the influences of magnetic fields on FOS-forming enzymes.

Taking into account the impact of other physical treatments, such as ultrasound and pulsed electric fields, on the enzyme kinetic parameters, similar qualitive increase of enzyme affinity was verified, as evidenced by a decrease on *K_m_* value of order of 13% for a commercial proteolytic enzyme submitted to ultrasound treatment [33]. On the other side, there was an opposite trend, i.e., an increase in *K_m_* values, after the exposure to endogenous ascorbate oxidase in carrot purée at low (<5 kJ kg^−1^) and medium (10–60 kJ kg^−1^) intensity [34].

## 4. Conclusions

The mixture of wheat and soy bran was more suitable for FFase production by *Aspergillus tamarii* Kita UCP 1279 than the medium composed only of wheat bran. The SSF conditions were investigated and the higher FFase activity (204.31 U mL^−1^) was obtained using 3 g of substrate, 20% of soy bran in medium and 15% of sucrose. The FFase produced presents optimum reaction conditions at 60 °C and pH range 5.0–7.0. The thermostability of the biocatalyst was evaluated through kinetic and thermodynamic study, and the FFase presents high stability at 50 °C, a temperature generally used for FOS production, evidenced by *t*_1/2_ and *D*-values of these results, confirmed by the thermodynamic parameters. The impact of a static magnetic field on FFase kinetic constants was positive due the increase in affinity of the enzyme by substrate, which was an indicator for further developments of industrial FOS production processes with the presence of a magnetic field.

## Figures and Tables

**Figure 1 biotech-12-00021-f001:**
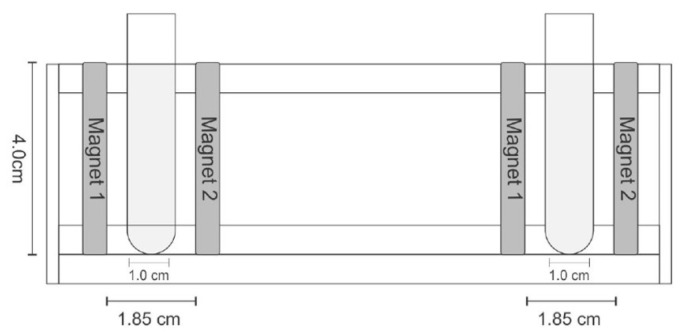
Magnets arrangement and tubes on support, with frontal view of support of tubes and spatial measurements.

**Figure 2 biotech-12-00021-f002:**
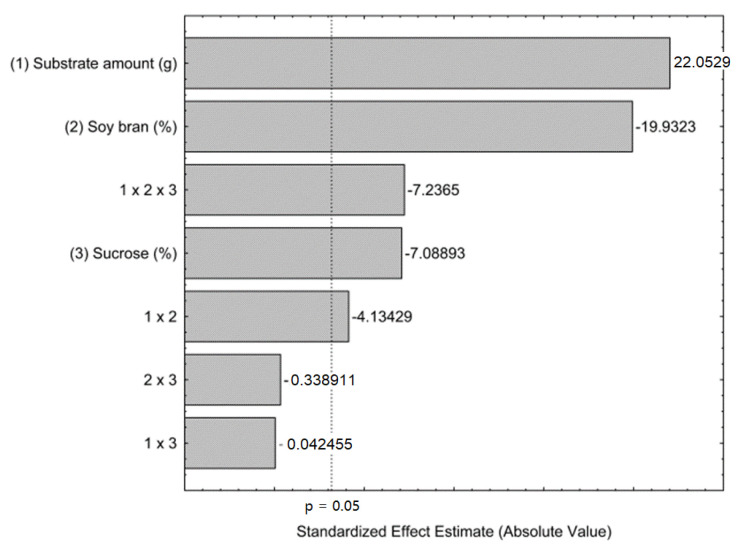
Pareto chart for effects of substrate amount, proportion of soy bran and sucrose concentration on the FFase production by *A. tamarii* Kita UCP 1279 using a mixture of wheat and soy bran as substrate through solid-state cultivation at 30 °C during 72 h.

**Figure 3 biotech-12-00021-f003:**
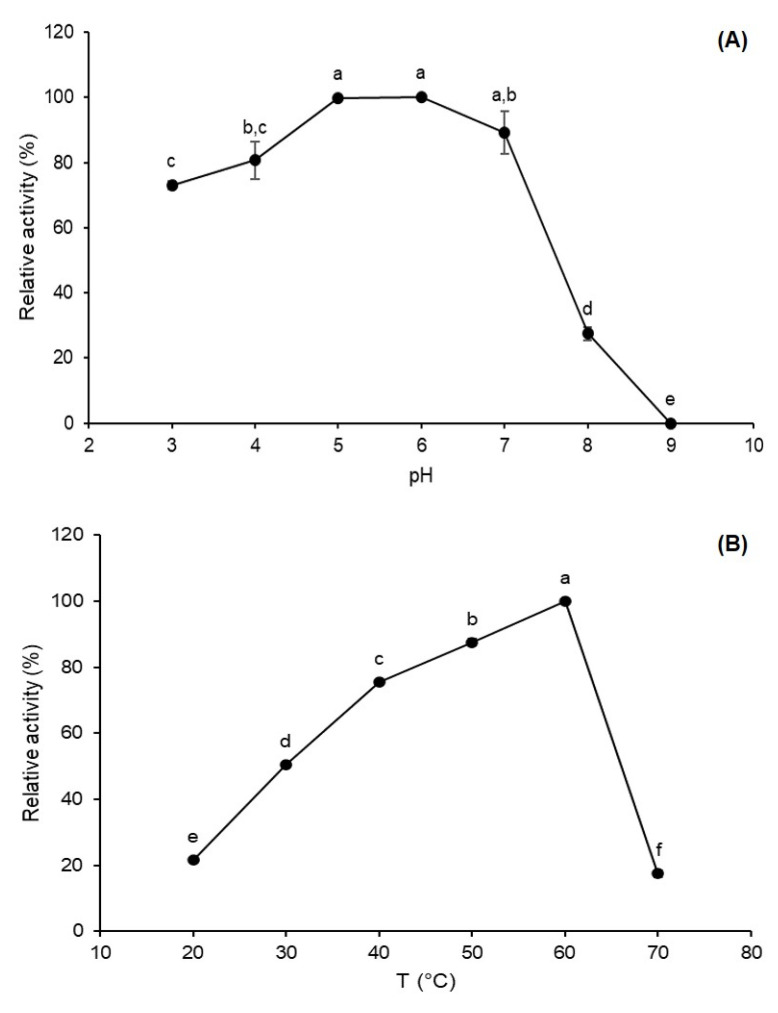
Effect of pH (**A**) and temperature (**B**) on FFase from *A. tamarii* Kita UCP 1279 obtained by solid-state fermentation using a mixture of wheat and soy bran as substrate. Different letters (a–f) indicate significant differences (*p* < 0.05).

**Figure 4 biotech-12-00021-f004:**
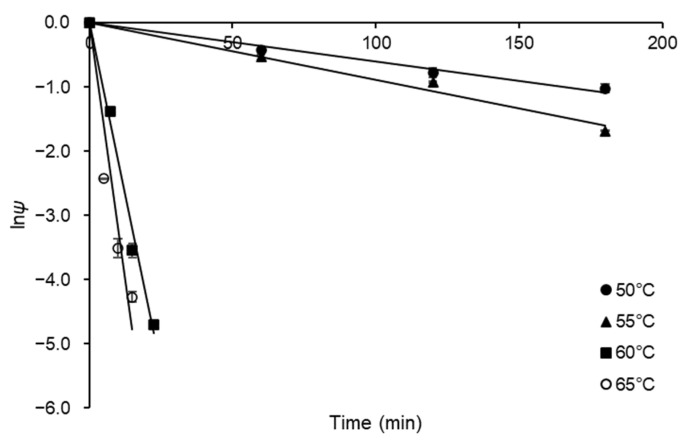
Semi-log plots of thermal denaturation of FFase from *A. tamarii* Kita UCP 1279 performed at pH 5.0 and using an initial substrate (sucrose) concentration of 60% (*w v*^−1^).

**Figure 5 biotech-12-00021-f005:**
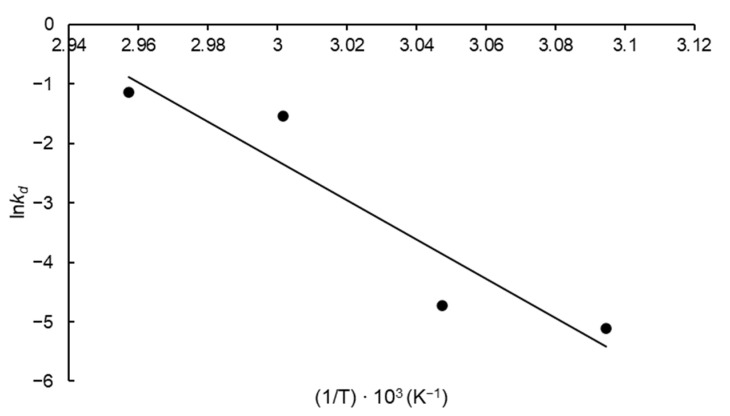
Arrhenius-type plot to estimate the thermal inactivation thermodynamic parameters of FFase from *A. tamarii* Kita UCP 1279 produced by solid-state fermentation using a mixture of wheat and soy bran as substrate (*R*^2^ = 0.88).

**Figure 6 biotech-12-00021-f006:**
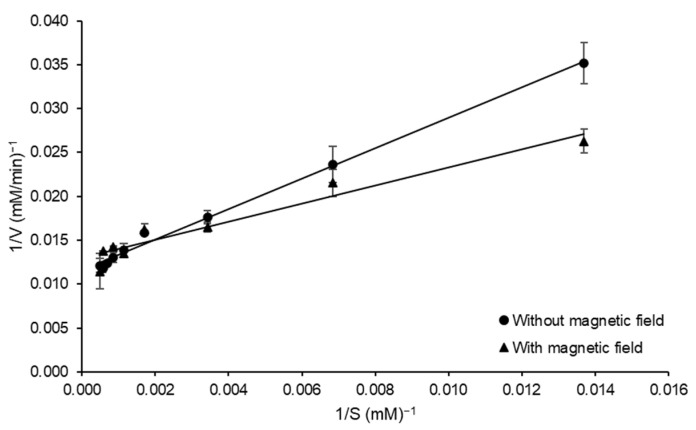
Lineweaver Burk plot of the initial rate of sucrose hydrolysis by FFase from *A. tamarii* Kita UCP 1279 1279 in the presence and absence of a magnetic field during catalyzed reaction versus sucrose concentration for the estimation of the kinetic constants at pH 5.0 and 55 °C.

**Table 1 biotech-12-00021-t001:** Experimental conditions and results of 2^3^ full factorial design used for evaluation of the production of FFase by *A. tamarii* Kita UCP1279, using a mixture of wheat and soy bran as substrate through solid-state cultivation.

Run	Substrate Amount (g)	Soy Bran (%)	Sucrose (%)	FFase Activity (U mL^−1^)
1	3	80	25	123.90
2	7	80	25	202.47
3	3	20	25	29.88
4	7	20	25	134.15
5	3	80	15	66.14
6	7	80	15	204.31
7	3	20	15	29.27
8	7	20	15	73.23
9 (C)	5	50	20	124.27
10 (C)	5	50	20	119.34
11 (C)	5	50	20	119.41
12 (C)	5	50	20	110.23

C—Central points.

**Table 2 biotech-12-00021-t002:** Kinetic and thermodynamic parameters for thermal denaturation of FFase from *Aspergillus tamarii* Kita UCP 1279.

^a^ T (°C)	^b^*k_d_* (min^−1^)	*R* ^2^	^c^*t*_1/2_ (min)	^d^*D*-Value (min)	^e^ Z (°C)	^f^*E***_d_* (kJ mol^−1^)	^g^ Δ*G***_d_* (kJ mol^−1^)	^h^ Δ*H***_d_* (kJ mol^−1^)	^i^ Δ*S*^*^*_d_* (J mol^−1^ K^−1^)
50	0.006	0.993	115.52	383.76	7.61	274.29	104.10	271.60	518.33
55	0.009	0.993	77.88	258.72			104.68	271.56	508.55
60	0.215	0.995	3.22	10.71			97.50	271.52	522.36
65	0.318	0.971	2.16	7.19			97.88	271.48	513.36

^a^ Temperature; ^b^ First-order rate constant; ^c^ Half-life; ^d^ Decimal reduction time; ^e^ Thermal resistance constant; ^f^ Activation energy of enzyme denaturation; ^g^ Activation Gibbs free energy; ^h^ Activation enthalpy and ^i^ Activation entropy.

## Data Availability

Raw data are available upon request.

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
