# Peer review of "Production, Kinetic/Thermodynamic Study, and Evaluation of the Influence of Static Magnetic Field on Kinetic Parameters of β-Fructofuranosidase from Aspergillus tamarii Kita UCP 1279 Produced by Solid-State Fermentation"

_biotech, 2023, doi:10.3390/biotech12010021_

Round 1

Reviewer 1 Report

The authors study the production of an FFase by fermentation in a solid medium followed by a characterization of the enzymatic extract (pH and optimum temperature and thermal stability) and the influence of magnetic fields on enzymes and the sucrose hydrolysis reaction. This study is interesting in the context of improving sucrose transformation processes for applications in the food industry. However, some elements remain to be further investigated in the studies carried out.

Intro part:

The introduction does not sufficiently describe the role of electromagnetic waves on enzymatic catalysis with specific examples that demonstrate this phenomenon + what are the applications in the food sector or even other sectors?

R&D part:

What is the specific activity of FFase given that the enzyme is not purified before use? it will be necessary to determine the amount of total protein to determine this specific activity.

What guarantees that the measured enzyme activity comes from FFase? Have the authors determined the other enzymatic activities present in the medium during solid fermentation?

Information is missing in all the figures: the legends of the figures must give the main conditions used in the experiments. The graphs in Figure 3 must show the evolution of the controls without enzyme to justify the action of the enzyme.

What means "radical pair recombination"? what kinds of molecules are involved and what are the mechanisms? give an hypothesis for the reaction studied here. What is the comparison and position of these results with those obtained with pulsed electric fields or thermal treatment or ultrasounds?

Author Response

ANSWERS TO THE SUGGESTIONS OF REVIEWER 1

The authors study the production of an FFase by fermentation in a solid medium followed by a characterization of the enzymatic extract (pH and optimum temperature and thermal stability) and the influence of magnetic fields on enzymes and the sucrose hydrolysis reaction. This study is interesting in the context of improving sucrose transformation processes for applications in the food industry. However, some elements remain to be further investigated in the studies carried out.

Intro part:

The introduction does not sufficiently describe the role of electromagnetic waves on enzymatic catalysis with specific examples that demonstrate this phenomenon + what are the applications in the food sector or even other sectors?

Answer: As indicated by the reviewer, a brief description of positive magnetic impact on bioprocess was added to introduction (Lines 80-82). Moreover, was added examples of studies involving the structural modifications from the exposure of magnetic field in enzymes of industrial and clinical interest (Lines 84-85).

R&D part:

What is the specific activity of FFase given that the enzyme is not purified before use? it will be necessary to determine the amount of total protein to determine this specific activity.

Answer: As the manuscript was not focused on the purification of the FFase crude extract, we did not present the specific activity results. This parameter was calculated in the beginning of study only in the condition used in the central points (5 g of substrate, 50% of soy bran and 20% of sucrose concentration) by the ratio of volumetric activity (118.31 U mL-1) and the protein content (0.58 mg mL-1), determined according to Bradford method, resulting in a specific activity of 203.98 U mg-1.

What guarantees that the measured enzyme activity comes from FFase? Have the authors determined the other enzymatic activities present in the medium during solid fermentation?

Answer: In the present study, we opted by evaluate the FFase crude extract in terms of the kinetic and thermodynamic parameters, in special observing the influence of magnetic field on kinetic parameters. However, other study performed by research group colleagues focused on the enzyme purification produced by the same fungal strain and was confirmed that enzyme can be identified as a β-fructofuranosidase (FFase), the reference of mentioned paper follows below:

Batista, J.M. da S.; Pedrosa Brandão-Costa, R.M.; Barbosa Cardoso, K.B.; Nascimento, T.P.; Albuquerque, W.W.C.; Carneiro da Cunha, M.N.; Porto, C.S.; Bezerra, R.P.; Figueiredo Porto, A.L. Biotechnological purification of a β-fructofuranosidase (β-FFase) from Aspergillus tamarii Kita: Aqueous two-phase system (PEG/Citrate) and biochemical characterization. Biocatal. Agric. Biotechnol. 2021, 35, 102070, doi:10.1016/j.bcab.2021.102070.

The crude extract was evaluated in terms of other enzymatic extracts and was detected protease activity, probably due the high protein content of wheat and soy bran, being a secondary enzymatic activity not mentioned in the manuscript.

Information is missing in all the figures: the legends of the figures must give the main conditions used in the experiments. The graphs in Figure 3 must show the evolution of the controls without enzyme to justify the action of the enzyme.

Answer: As requested by the reviewer, the description of experimental conditions in the figure captions were improved, especially in the Figures 2, 3, 4, 5 and 6. As the evaluation of the effects of pH and temperature on FFase hydrolytic activity was performed using substate (sucrose) with high purity degree, the enzymatic activity obtained in each condition evaluate is only from the FFase, so there is no need to use controls without enzyme just the reaction blank used in spectrophotometric reaction, whose absorbance results are not usually presented in graphs.

What means "radical pair recombination"? what kinds of molecules are involved and what are the mechanisms? give an hypothesis for the reaction studied here. What is the comparison and position of these results with those obtained with pulsed electric fields or thermal treatment or ultrasounds?

Answer: The radical pair recombination corresponds to a model for describing the effect of magnetic field on the enzyme kinetics initially proposed by Eichwald and Walleczek (1996). These authors extend to Michaelis-Menten reaction the model preexistent to photosynthetic reaction, indicating that in the intermediate enzyme-substrate complex has a spin-correlated radical pair. Therefore, the magnitude of effect o magnetic field can be determined by the ratio between radical pair lifetime and the rate of magnetic field-sensitive intersystem. In the case of the sucrose hydrolysis reaction, until the moment, there are no reports in the scientific literature regarding to elucidate the mechanism of radical pair recombination or propose a new explanation for hydrolysis reaction. To best of our knowledge, the studies published focused on the elucidation evaluated only to lyases and redox enzymes. Taking account, the impact of other physical treatments as ultrasound and pulsed electric fields on the enzyme kinetic parameters was verified that similar qualitive increase of enzyme affinity evidenced by a decrease on Km value for a commercial proteolytic enzyme submitted to ultrasound and an opposite trend, i.e., an increase in Km values, after the exposition of ascorbate oxidase in carrot purée at low (< 5 kJ kg-1) and medium (10-60 kJ kg-1) intensity. The references of mentioned papers follow below, moreover, a brief discussion of involving these studies were added to manuscript (Lines 328-333).

Jia, J.; Ma, H.; Zhao, W.; Wang, Z.; Tian, W.; Luo, L.; He, R. The use of ultrasound for enzymatic preparation of ACE-inhibitory peptides from wheat germ protein. Food Chem. 2010, 119, 336–342, doi:10.1016/j.foodchem.2009.06.036.

Leong, S.Y.; Oey, I. Effect of pulsed electric field treatment on enzyme kinetics and thermostability of endogenous ascorbic acid oxidase in carrots (Daucus carota cv. Nantes). Food Chem. 2014, 146, 538–547, doi:10.1016/j.foodchem.2013.09.096.

Reviewer 2 Report

20 February 2023

Manuscript ID: biotech-2222874

Title: Production, kinetic/thermodynamic study, and evaluation of the effect of static magnetic field on kinetic parameters of β-fructofuranosidase from Aspergillus tamarii Kita UCP 1269 produced by solid-state fermentation

Author: Rodrigo Lira De Oliveira, Aldeci FrançaAraújo Dos Santos, Bianca Alencar Cardoso, Thayanne Samille da Silva Santos, Galba Maria de Campos-Takaki, Tatiana Souza Porto, Camila Souza Porto

The first objective of the work, solid-state fermentation conditions forβ-fructofuranosidase production, with 23- full factorial design was presented satisfactorily. The second objective of the work, the effect of pH and temperature on sucrose hydrolysis catalyzed by β-fructofuranosidase and kinetic/thermodynamic parameters of thermal inactivation was comprehensively presented. Presented the impact of the exposition of static magnetic field on kinetic constants (Km and Vmax) of β-fructofuranosidase provides a better understanding of enzymatic reaction.

 I present some editorial errors below:

1.      In line 109, please explain “centrifugedfor 20 min at 3,000 x g”

2.      In line 114 correct “23” on “23

3.      In line 149 in Eq. (3) consider the description “D” not “D – value”

4.       In lines 160 and 161 correct units “J.s” and “J.K-1

5.      Figure 3. What means “a, b, c, d, e, f” on plots A) B)? 

6.      Correct the writing of the description unit ( °C) after space, for example “30 °C”

7.      Table 2 and line 257 please  correct the description “E*d” as the description in line 153

8.      Reference. Correct Ref. 7, 18, 22, 

9.      In Ref. 28  write “Mucor subtilissimus” in italic

Conclusion: I recommend this manuscript to publish in BioTech by correct carefully the references.

Author Response

ANSWERS TO THE SUGGESTIONS OF REVIEWER 2

The first objective of the work, solid-state fermentation conditions forβ-fructofuranosidase production, with 23- full factorial design was presented satisfactorily. The second objective of the work, the effect of pH and temperature on sucrose hydrolysis catalyzed by β-fructofuranosidase and kinetic/thermodynamic parameters of thermal inactivation was comprehensively presented. Presented the impact of the exposition of static magnetic field on kinetic constants (Km and Vmax) of β-fructofuranosidase provides a better understanding of enzymatic reaction.

I present some editorial errors below:

  1. In line 109, please explain “centrifuged for 20 min at 3,000 x g”

Answer: The centrifugation operation was performed to remove the solids present on crude extract after the filtration through cheesecloth. The description of the centrifugation process were improved in the text (Lines 114-115).

  1. In line 114 correct “23” on “23

Answer: As indicated by the reviewer, the correction was performed.

  1. In line 149 in Eq. (3) consider the description “D” not “D – value”

Answer: As suggested by the reviewer, in the Equation (3) we opted to use D for represent the decimal reduction time.

  1. In lines 160 and 161 correct units “J.s” and “J.K-1

Answer: As indicated by the reviewer, the units of the Planck and Boltzmann constants were corrected.

  1. Figure 3. What means “a, b, c, d, e, f” on plots A) B)? 

Answer: The different letters correspond to statistical different samples determined according to Tukey’s test as mentioned in Section 2.4. To facilitate the reader’s understanding, this information was indicated in the caption of the figure (Lines 232-233).

  1. Correct the writing of the description unit ( °C) after space, for example “30 °C”

Answer: As indicated by the reviewer, were verify all mention of unit (°C) in the text and the corrections were performed.

  1. Table 2 and line 257 please correct the description “E*d” as the description in line 153

Answer: As indicated by the reviewer, the description of the parameter activation energy of enzyme denaturation was corrected and standardized in the text.

  1. Reference. Correct Ref. 7, 18, 22, 

Answer: As requested by the reviewer, the indicated references were verified and adjusted.

  1. In Ref. 28  write “Mucor subtilissimus” in italic

Answer: As indicated by the reviewer, the correction was performed.

Conclusion: I recommend this manuscript to publish in BioTech by correct carefully the references.

Reviewer 3 Report

In this study, the authors investigated the production of β-fructofuranosidases (FFases) by Aspergillus tamarii Kita UCP 1279 using solid-state fermentation with a mixture of wheat and soy brans as substrate. The optimal pH and temperature for FFase activity were found to be 5.0-7.0 and 60°C, respectively. Additionally, the study found that a static magnetic field with a 1450 G magnetic flux density had a positive impact on FFase kinetic parameters, increasing the enzyme's affinity for substrate. These results suggest that FFases from Aspergillus tamarii Kita UCP 1279 have suitable characteristics for use in the food industry, and the magnetic field’s great potential for bioprocesses. Overall, the discovery of this study is interesting which can attract the readers in the field. However, the following minor concerns should be addressed before the paper being accepted.

Minor:

1. Please provide high resolution figure 1.

2. Reference 1 has the first author’s name as “de Oliveira, R.L.” while the same name appears as “ Oliveira, R.L. de” for the reference 7. Other researchers’ names have the same issue. Please make corrections according to the journal’s requirement and double check all the other references.

Author Response

ANSWERS TO THE SUGGESTIONS OF REVIEWER 3

In this study, the authors investigated the production of β-fructofuranosidases (FFases) by Aspergillus tamarii Kita UCP 1279 using solid-state fermentation with a mixture of wheat and soy brans as substrate. The optimal pH and temperature for FFase activity were found to be 5.0-7.0 and 60°C, respectively. Additionally, the study found that a static magnetic field with a 1450 G magnetic flux density had a positive impact on FFase kinetic parameters, increasing the enzyme's affinity for substrate. These results suggest that FFases from Aspergillus tamarii Kita UCP 1279 have suitable characteristics for use in the food industry, and the magnetic field’s great potential for bioprocesses. Overall, the discovery of this study is interesting which can attract the readers in the field. However, the following minor concerns should be addressed before the paper being accepted.

Minor:

  1. Please provide high resolution figure 1.

Answer: As requested by the author, the resolution of Figure 1 was improved.

  1. Reference 1 has the first author’s name as “de Oliveira, R.L.” while the same name appears as “Oliveira, R.L. de” for the reference 7. Other researchers’ names have the same issue. Please make corrections according to the journal’s requirement and double check all the other references.

Answer: As indicated by the reviewer, the mentioned reference was adjusted and as well as other reference in the same situation.

Round 2

Reviewer 1 Report

The authors answered the questions with interest and provided additional information

I agree for the article to be published in BioTech